

# The chloroplast genome inheritance pattern of the Deli-Nigerian prospection material (NPM) × Yangambi population of *Elaeis guineensis* Jacq

Nurul Shakina Mohd Talkah[1], Nur Afieqah Kaz Abdul Aziz[2], Muhammad Farid Abdul Rahim[3], Nurul Fatiha Farhana Hanafi[3], Mohd Azinuddin Ahmad Mokhtar[3] and Ahmad Sofiman Othman[1,2]

[1] School of Biological Sciences, Universiti Sains Malaysia, Minden, Pulau Pinang, Malaysia
[2] Centre of Chemical Biology, Universiti Sains Malaysia, Bayan Baru, Pulau Pinang, Malaysia
[3] Plant Breeding Unit, FGV R&D Sdn Bhd, Bandar Pusat Jengka, Pahang, Malaysia

Corresponding author
Ahmad Sofiman Othman,
sofiman@usm.my

## ABSTRACT

**Background**. The chloroplast genome has the potential to be genetically engineered to enhance the agronomic value of major crops. As a crop plant with major economic value, it is important to understand every aspect of the genetic inheritance pattern among *Elaeis guineensis* individuals to ensure the traceability of agronomic traits.

**Methods**. Two parental *E. guineensis* individuals and 23 of their $F_1$ progenies were collected and sequenced using the next-generation sequencing (NGS) technique on the Illumina platform. Chloroplast genomes were assembled *de novo* from the cleaned raw reads and aligned to check for variations. The sequences were compared and analyzed with programming language scripting and relevant bioinformatic softwares. Simple sequence repeat (SSR) loci were determined from the chloroplast genome.

**Results**. The chloroplast genome assembly resulted in 156,983 bp, 156,988 bp, 156,982 bp, and 156,984 bp. The gene content and arrangements were consistent with the reference genome published in the GenBank database. Seventy-eight SSRs were detected in the chloroplast genome, with most located in the intergenic spacer region. The chloroplast genomes of 17 $F_1$ progenies were exact copies of the maternal parent, while six individuals showed a single variation in the sequence. Despite the significant variation displayed by the male parent, all the nucleotide variations were synonymous. This study show highly conserve gene content and sequence in *Elaeis guineensis* chloroplast genomes. Maternal inheritance of chloroplast genome among $F_1$ progenies are robust with a low possibility of mutations over generations. The findings in this study can enlighten inheritance pattern of *Elaeis guineensis* chloroplast genome especially among crops' scientists who consider using chloroplast genome for agronomic trait modifications.

## INTRODUCTION

The oil palm (*Elaeis guineensis* Jacq.) is one of the most economically productive oil crops and produces the most versatile vegetable oil in the world. Global palm oil production is continuously increasing, with 2020 global annual production reaching 74.7 million metric tonnes (*FAO, 2022*). The crop thrives in the tropical environment, specifically, in a belt spanning the equator, with the major producers being Indonesia followed by Malaysia (*Abubakar & Ishak, 2022*). The palm oil demand will increase with the growing world population and in 2050 is expected to reach 240 million tonnes (*Corley, 2009*).

One of the ways to sustainably produce more palm oil is *via* plant breeding. For example, the discovery of shell gene inheritance led to significantly greater yields (*Beirnaert & Vanderweyen, 1941*) through the development of the *tenera* palm, a hybrid of the thick-shell *dura* and the shell-less *pisifera*, which produces a thin-shell fruit that contains 30% more oil than the *dura* variety (*Seng et al., 2011*). Since then, the *tenera* palm has been widely planted by plantations. Thus, continuous yield improvement through the genetic exploitation of various lineages is necessary for developing new varieties, especially in preparing the industry for climate change and improving agronomic traits. Currently, commercial seed production utilizes lineages from Algemeene Vereeniging van Rubberplanters ter Oostkust van Sumatra (AVROS), including dumpy AVROS, dumpy Yangambi, dumpy Yangambi x AVROS, Calabar, and Ekona (*Rajanaidu et al., 2013*), and Felda Global Ventures Holdings Berhad (FGV), with a focus on exploiting the Yangambi background.

Besides conventional breeding, various investigations have successfully enhanced plants' agronomic traits by genetically modifying the nuclear genome (*Řepková, 2010*; *An et al., 2022*). Despite their frequent use, transgene expression and hereditary agronomic characteristics are challenging to regulate because of their nuclear biparental characteristics. There is also a risk that the transgene may spread among the plants' wild relatives (*Řepková, 2010*) through pollen dispersion. The chloroplast, one of the organelles present in the plant cell, harbors an independent genome that is small in size, and has mostly been reported to be uniparentally inherited from maternal parents, and thus has a lower risk of being spread through pollen (*Daniell, 2007*; *Birky, 2008*; *Park et al., 2021*; *An et al., 2022*), making it a suitable genome for inheritance studies and genetic engineering. Examples of genetic engineering performed on the chloroplast genome to improve agronomic traits include herbicide detoxification (*Bansal & Saha, 2012*), resistance to stress, nutritional value enhancement, biopharmaceuticals, and vaccine development (*Daniell et al., 2016*).

The chloroplast genome in angiosperms is quite conserved and is made up of a typical quadripartite structure, with large single-copy (LSC) and small single-copy (SSC) regions, as well as two copies of inverted repeats (IRs) separating the LSC and SSC. The genome size is usually between 120 and 170 kb in length. Previously, the generation of *E. guineensis* chloroplast genome data has been performed in a small number of works focusing on genome characterizations and the deep phylogenomic analysis of palm species. Based on the first reference genome (NC_017602) (*Uthaipaisanwong et al., 2012*), *E. guineensis* chloroplast genome has 156,973 bp in size with 112 unique genes. Comparative chloroplast genomes analysis between *E. guineensis* and date palm species revealed chloroplast genomes

size differences (1,489 bp). However, their chloroplast genome structural characteristics and gene contents are similar. In another study (*Yao et al., 2023*), *E. guineensis* chloroplast genome was utilized for deep phylogenomic study within Arecaceae species. In addition, the chloroplast genome sequence (ON248756) shows a few gaps (N) present in the genome alignment. Both studies (*Uthaipaisanwong et al., 2012*; *Yao et al., 2023*) showed the gene content and arrangement in chloroplast genome are highly conserved in the palm family. Comparative analysis of chloroplast genomes was done between three different palm species from Brazil (*Francisconi et al., 2022*). This study showed these three species' nucleotide sequences are significantly varied from each other in term of insertion/deletion (InDels) and single nucleotide polymorphisms (SNPs). Although there are variations in the coding-regions among species involved, the region with high polymorphic diversity is the intergenic regions where most of the SNPs are found.

In most angiosperm, chloroplast genome are maternally inherited. Nonetheless, biparental inheritance of the chloroplast genome in angiosperms has been reported several times (*Mason, Holsinger & Jansen, 1994*; *Hansen et al., 2007*; *Barnard-Kubow, McCoy & Galloway, 2017*). Previous investigations on *E. guineensis* chloroplast genome did not highlight in detail the chloroplast genome inheritance pattern among *E. guineensis* individuals and their parental individuals. It is therefore pertinent to study the inheritance pattern of the chloroplast genome among *E. guineensis* individuals, especially in those of parental and $F_1$ progenies in a breeding programme.

In this study, the parents and progenies of *E. guineensis* in an ongoing breeding research programme by FGV were used. Based on simple sequence repeat (SSR) analysis performed by FGV, a few individuals in the progeny showed ambiguous relationships with their putative parents. As such, this article aims to clarify inheritance patterns among Deli-Nigerian prospection material (NPM) × Yangambi, specifically the GB33 population (J4-25 × ML-161), by comparing their chloroplast genome structure to that of their putative parents. By observing and comparing nucleotide variations between maternal and paternal parent of these $F_1$ progenies, any regions that signals possible mutation could be highlighted for future reference in breeding program.

## MATERIALS & METHODS

### Plant materials, deoxyribonucleic acid extraction and sequencing

The study involved 25 samples of the GB33 population, planted at the FGV trial plot, Pusat Penyelidikan Pertanian Tun Razak, Jengka, Pahang. The 25 individual palms selected in this article consisted of the parental samples, J4-25 (*dura*, Deli-NPM), ML-161 (*pisifera*, Yangambi), and 23 samples of their $F_1$ progenies. Young leaf samples from each $F_1$ individuals and of parents were collected, dried in silica gel and stored at −20 °C until deoxyribonucleic acid (DNA) extraction. The leaf samples were ground to a fine powder form utilizing liquid nitrogen. The DNeasy Plant Mini Kit (Qiagen, Hilden, Germany) was used to extract and purify DNA from the leaf samples. The success of genomic DNA extraction was evaluated on 1.0% (weight/volume) agarose gels in 0.5 × Tris-Borate-EDTA (TBE) buffer. Genomic DNA quantification and purity assessment were measured with a

NanoDrop 2000 spectrophotometer (Nanodrop, Wilmington, DE, USA). Selection criteria for samples include a concentration of $\geq$ 5.0 ng/µl, the 260/280 ratio within the range of 1.80 to 2.00, and the 260/230 ratio within the range of 1.95 to 2.22. Samples meeting these criteria were deemed free from protein or other contaminants. DNA quantities of $\geq$ 50 ng were considered sufficient and suitable for subsequent sequencing protocol. Short-read paired-end (125 bp) sequencing was performed *via* the Illumina HiSeq Platform (Shanghai Biozeron Biotechnology Co., Ltd). The sequencing depth was 10X whole-genome coverage for all $F_1$ progenies and 40X whole-genome coverage for both ML-161 and J4-25 individuals.

## Genome assembly and annotations

Upon assembly, the raw reads with low quality base reads were checked for any contamination and low-quality base using FastQC. Any problematic regions were cut by Trimmomatic software (*Bolger, Lohse & Usadel, 2014*). The cleaned raw reads were deposited in the sequence read archive (SRA) of the National Center for Biotechnology Information (NCBI). Raw reads for each sample were independently assembled in *de novo* mode using NOVOPlasty (*Dierckxsens, Mardulyn & Smits, 2017*) version 4.2.1 on a Linux workstation. Any ambiguous regions or nucleotides from the draft assemblies were resolved by re-assembling raw reads using GetOrganelle version 1.7.5 (*Jin et al., 2020*). Although the assemblies were performed in *de novo* mode, the reference genome from the GenBank was used to ensure genome orientation. Genome annotations were performed using GeSeq (*Tillich et al., 2017*) (https://chlorobox.mpimp-golm.mpg.de/geseq.html) and CPGAVAS2 (*Shi et al., 2019*) with the existing *E. guineensis* chloroplast genome (NC_017602) as a reference sequence. Open-reading frames for protein-coding genes were confirmed by NCBI Blast (*Boratyn et al., 2013*). The tRNA genes were confirmed using tRNA-SE. Annotation tables were prepared using GB2Sequin (*Lehwark & Greiner, 2019*) and the complete chloroplast genomes were deposited in the GenBank through NCBI BankIt. Chloroplast genome visualization was modified from figure generated by OGDRAW.

## Single nucleotide polymorphism detection and validation

The sequences were edited to homologous positions using MEGA-X (*Kumar et al., 2018*) before being aligned by MAFFT version 7.511 (*Katoh & Standley, 2013*). Aligned sequences were manually viewed using Jalview version 2.11.2.6 (*Waterhouse et al., 2005*) to search for any available single nucleotide polymorphisms (SNPs) and InDels (nucleotide insertion and deletion). The variations in the alignment were confirmed using DnaSP v6 (*Rozas et al., 2017*). Summary of nucleotide variations were visualized using python script with J425 as reference sequence. Nucleotides that appeared as SNPs and insertion in the assembled chloroplast genomes were validated using coverage analysis. The genome sequences were mapped back to their raw reads using a Burrows-Wheeler aligner (*Li & Durbin, 2009*) and the read depth for each nucleotide position was retrieved using SamTools (*Li et al., 2009*).

## Haplotype and SSR detection

Haplotype analysis was conducted using DnaSP v6 to identify different types of chloroplast genome sequences in this study. Analysis results were then exported in nexus format using the alignment of all samples along with the *E. guineensis* reference genome (NC_017602).

PopART software (*Leigh & Bryant, 2015*) was used to build a TCS network diagram. SSR were detected from the J4-25 chloroplast genome using MISA (*Beier et al., 2017*) with minimum repeat sizes of 10 for mononucleotide, five for dinucleotide, four for trinucleotide and three for tetranucleotide, pentanucleotide and hexanucleotide, respectively. Finally, REPuter (*Kurtz et al., 2001*) program was utilized to characterize repeat types of J4-25 chloroplast genome with parameters of 30 minimal repeat size and 3 hamming distance value.

## RESULTS

### Raw reads and assembled chloroplast genome features

The cleaned reads for all samples comprised of 126,179,838–398,341,772 reads with 18,903,632,149–59,677,199,365 bases (Table S1). The Q30 for the cleaned reads were between 91.6%–96.89%. All cleaned reads were deposited in the SRA of NCBI servers (Table S1). Chloroplast genome assemblies resulted in 156,983, 156,984 and 156,988 bp lengths. All chloroplast genomes possessed a quadripartite structure which are large single-copy (LSC), small single-copy (SSC) and a pair of inverted repeats (IR)s, IRa and IRb. J4-25 and 17 other $F_1$ individuals showed a total of 156,983 bp in length with 85,194 bp of LSC), 17,643 bp of SSC and 27,073 bp of IR. GB33-41 and GB33-46 individuals showed 156,984 bp in length for the cp genome, while GB33-17 showed 156,892 bp. While for ML-161, the total length of the cp genome was 156,988 bp (85,196 bp LSC, 17,644 bp SSC and 27,074 bp IR). Guanine and cytosine (GC) content for all assembled chloroplast genomes is 37.4%. Due to the slight total length variations, the length of those regions also varies. Details on assembly are available in Table S2. The total number of genes annotated were similar in all chloroplast genomes, with 79 protein-coding genes, 29 tRNA genes and four rRNA genes. The number of annotated genes is also consistent with the reference genome uploaded in the GenBank (NC_017602).

### SNP and InDel characteristics

The variations detected from ML-161 were only 15 SNPs and nine InDels when compared to J4-25 (Table S3). Coverage depth for each nucleotide variation is listed in Table S3. For most of the $F_1$ progenies, the chloroplast genome was the exact copy of J4-25 except for six $F_1$ progenies (GB33-47, GB33-46, GB33-31, GB33-11, GB33-41 and GB33-17). The variations showed by the $F_1$ progenies with J4-25 contained only one nucleotide variation for each individual. Figure 1 shows variations among chloroplast genomes compared to the J4-25 individual which were further described in detail through Table S4. Figure 2 depicts the circular visualisation of the J4-25 chloroplast genome and nucleotide variations compared to other study samples. Although the number of variations between ML-161 and J4-25 are significant, Table 1 shows that all variations in the protein coding region are synonymous.

### Haplotype network and SSR analysis

There are eight haplotypes across 94 variation sites present in the final alignment with the GenBank reference genome. The individuals in the same haplotypes are listed in Table 2.
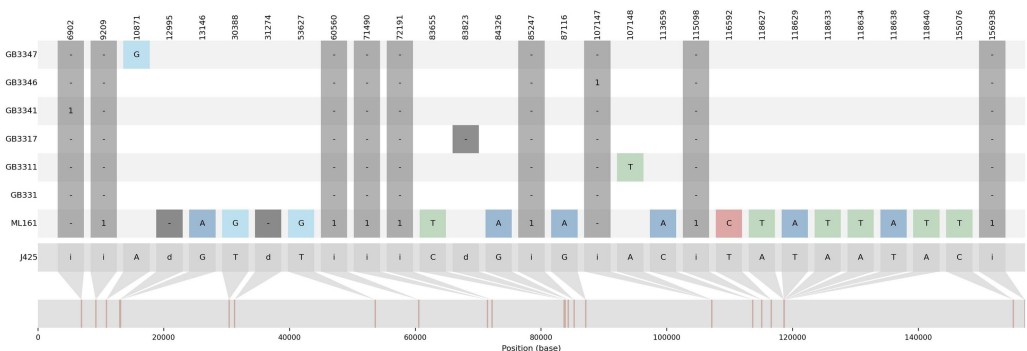

**Figure 1** Summary visualization of nucleotide variations in the sequences generated in this study compared to J4-25. J4-25 serve as reference genome on the bottom. The numbers on the top diagram are alignment position for each variation. The box in dark grey color are InDels variation with 'i' is insertion and 'd' is deletion. The number in the dark grey box shows the number of inserted nucleotide. For example, GB3341 has one insertion in position 6,902, ML161 has one deletion in position 12,995. Additionally, SNP variations were also specified in the diagram.

The TCS network among haplotypes can be seen in Fig. 3. Based on MISA output, 78 SSRs were detected from the J4-25 chloroplast genome with 45 mononucleotide, 18 dinucleotide, two trinucleotide, nine tetranucleotide, two pentanucleotide and one hexanucleotide. From these, 11 SSRs exist in the compound form. Figure 4A shows the distribution of SSRs in the chloroplast genome. Most of the SSRs detected are in the intergenic region. Table S5 is the MISA file generated based on J4-25 chloroplast genome. REPuter revealed a total of 67 repeats in the chloroplast genome, with 24 forward, 12 reverse, four complementary and 27 palindrome repeats. The result was summarized as a chart in Fig. 4B.

## DISCUSSION

This study revealed chloroplast genome inheritance patterns among 23 *E. guineensis tenera* individuals in a single population. The generation of *E. guineensis* chloroplast genome data has been performed in a small number of works focusing on genome characterizations (*Uthaipaisanwong et al., 2012*) and the deep phylogenomic analysis of palm species (*Yao et al., 2023*). However, these investigations did not highlight in detail the chloroplast genome inheritance pattern among *E. guineensis* individuals from parent individuals. Various analyses have shown that maternal inheritance of the chloroplast genome is particularly prevalent among angiosperms, for example, in cucumber (*Park et al., 2021*), spinach (*She et al., 2022*) and banana (*Fauré et al., 1994*). There is also evidence that the chloroplast genome from angiosperm plants can be inherited paternally and biparentally (*Shrestha et al., 2021*). In the present study, the paternal chloroplast genome (ML-161) displayed a more significant number of variations than the maternal one, J4-25. Based on coverage depth analysis, the reliability of these nucleotide sequences had a high confidence level (>100×) (Table S4). All variations in ML-161 were single-nucleotide except for one position in *ndh* D–*psa* C, which demonstrated two variations. Nucleotide variations, either

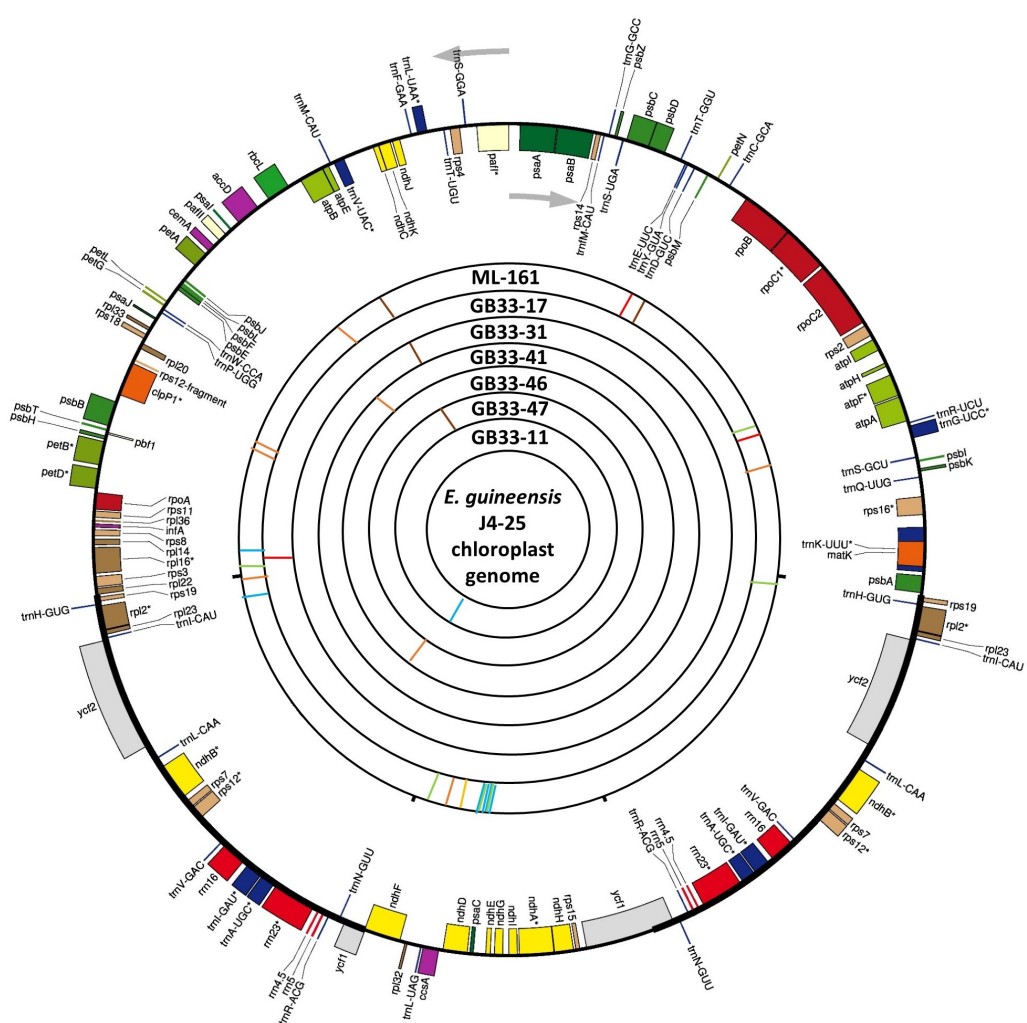

**Figure 2** Visualization of the circular structure of *E. guineensis* (J4-25) chloroplast genome. The arrows show the transcription direction of the genes. Genes positioned on the outside are transcribed counterclockwise, conversely, genes positioned on the inside are transcribed clockwise. The bands on the inside represent SNPs and InDels of different individuals in this study. Red: deletion; orange: Insertion; brown: SNP 'G'; green: SNP 'A'; blue: SNP 'T'; yellow: SNP 'C'.

single-nucleotide polymorphisms (SNPs) or insertion–deletion mutations (indels) between sequences, mostly occurred in the non-coding region (intron and intergenic spacer).

Despite being assembled independently in *de novo* mode, 17 $F_1$ progenies inherited the exact sequences of their maternal chloroplast genome (J4-25). Conversely, six of the $F_1$ progenies demonstrated a single variation in nucleotide position, all within intergenic regions. High sequence variations in the chloroplast intergenic regions have been consistently reported in many works (*Hamilton, 1999*; *Abdullah Shahzadi et al., 2019*; *Gichira et al., 2019*). The data generated from the present study confirm this earlier work, showing that intergenic regions have a higher evolutionary rate than the coding region.

**Table 1  Amino acid mutation in coding region among individuals in this study.**

| Protein coding gene | Sample involved | SNP | Amino acid mutation | Synonymous/ Non-synonymous |
| --- | --- | --- | --- | --- |
| *ndh* F | ML161 | 'G' to 'T' | None | Synonymous |
| *rps* 3 | ML161 | 'A' to 'G' | None | Synonymous |
| *rpl* 2 | ML161 | 'T' to 'C' | None | Synonymous |
| *ccs* A | ML161 | 'C' to 'T' | None | Synonymous |

**Table 2  Haplotypes among chloroplast genome sequences generated in this study and reference genome in GenBank.**

| Haplotypes | Sample individual(s) |
| --- | --- |
| Hap_1 | NC_017602 (Reference genome from GenBank) |
| Hap_2 | J4-25, GB331, GB332, GB333, GB334, GB335, GB336, GB339, GB3313, GB3319, GB3326, GB3327, GB3329, GB3332, GB3340, GB3344, GB3348, GB3351 |
| Hap_3 | ML-161 |
| Hap_4 | GB3311 |
| Hap_5 | GB3317 |
| Hap_6 | GB3331, GB3347 |
| Hap_7 | GB3341 |
| Hap_8 | GB3346 |

Although variations also occurred within the coding region, they were synonymous, which means that there were no amino acid changes in the transcription process.

Accessibility to genomics data in recent decades has demonstrated the existence of plastid sequence polymorphisms among different plant populations and genera (*Kim et al., 2015*; *Henriquez et al., 2020*; *Wei et al., 2021*). Plastid genome characteristics have been particularly useful in assisting evolutionary and diversity studies among interspecific individuals. Nevertheless, when variations exist among intraspecific individuals, different approaches or interpretations are required. Previous works have reported the occurrence of intraspecific variations in the chloroplast genomes of several crop plants: the 'Red Fuji' apple (*Miao et al., 2022*), the castor bean (*Muraguri et al., 2020*), red clover (*Li et al., 2019*) and Korean ginseng (*Jo et al., 2016*). As a perennial plant, *E. guineensis* has the potential to develop intra-individual genomic variation because of its ability to independently grow the same organ at different times (*Sun et al., 2019*). Nonetheless, this assumption is currently theoretical and requires further analysis.

Further assessment in haplotype analysis involves the reference genome available in the GenBank database. Although the variant is different, ML-161 and J4-25 clearly showed a much closer relationship compared to the reference genome. It is particularly important to keep track of genetic inheritance patterns of haplotypes in crop breeding pedigrees to support new genomic combinations with desired traits (*Varshney et al., 2021*). In jujube breeding, chloroplast genome haplotypes were previously used as a guide for the next breeding plan in order to produce fruits with highly desirable traits (*Hu et al., 2022*). For

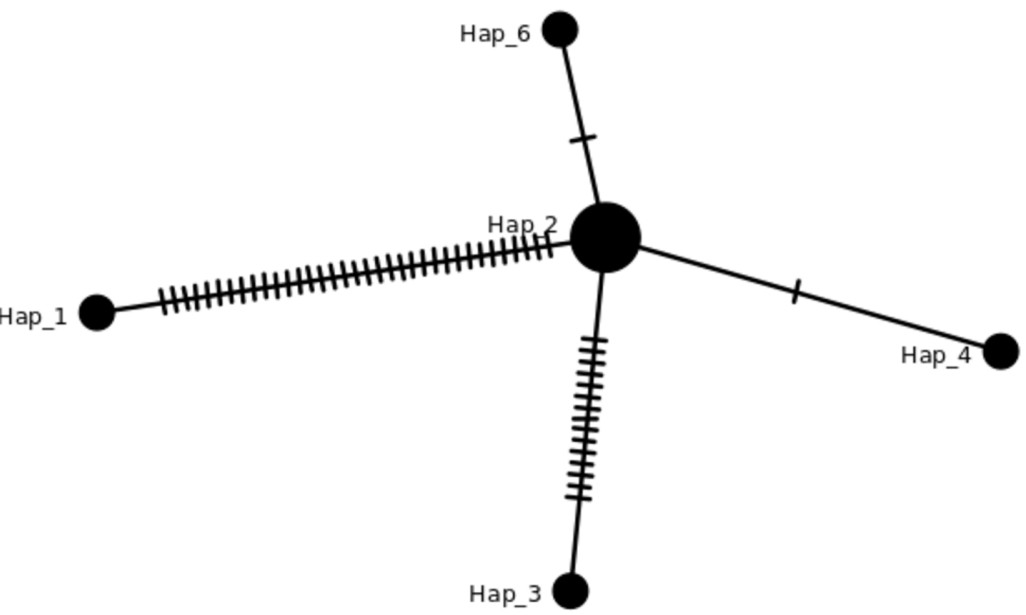

**Figure 3 TCS network of haplotypes in this study.** Haplotypes individuals with SNPs variation can be viewed in Table 2 and the details in Table S2. The hash lines between different haplotypes are nucleotide SNPs. Both Hap_6 (GB3331,GB3347) and Hap_4 (GB3311) showed one SNP, Hap_3 (ML-161) showed 15 SNPs, and Hap_1 (NC_017602) showed 35 SNPs compared to J4-25 sequence.

a well-studied crop plant such as *E. guineensis*, only certain pedigrees are chosen because they carry the desired traits for commercial seed production. Evidently, in *E. guineensis*, the maternal inheritance pattern is quite robust despite minimal variation compared to ML-161. The highly conserved sequence displayed by the $F_1$ progenies demonstrates that there is considerably high potential for transferring the genetic characteristics from the maternal unit to the progenies without mutation.

SSR sequences have been known to be very useful in conducting population studies. The polymorphisms in the chloroplast SSR (cpSSR) usually vary across species and loci (*Amiteye, 2021*). SSRs from the chloroplast genome are desirable because they are easy to isolate with polymorphic characteristics, particularly in the intergenic spacer region. A comparative analysis of plastid SSRs among palm trees recently showed that most of the detected SSRs are mononucleotides within the non-coding region (*Silveira et al., 2022*). Similarly, the SSRs observed in the present investigation were mostly from single copy intergenic spacer regions, with 36 being mononucleotide repeats. These SSRs have the potential to be developed as molecular markers especially for genotype discrimination purposes. Despite report on low discriminatory ability of *E. guineensis* restriction fragment length polymorphism (RFLP) markers from chloroplast region (*Jack, Dimitrijevic & Mayes, 1995*), the discriminatory ability of these SSR markers for this species can still being assessed in the future. Low diversity was expected among samples in this study because the plant samples employed were intraspecific, in addition to the clonal inheritance and non-recombinant nature of the chloroplast genome. All four repeat types (palindrome,

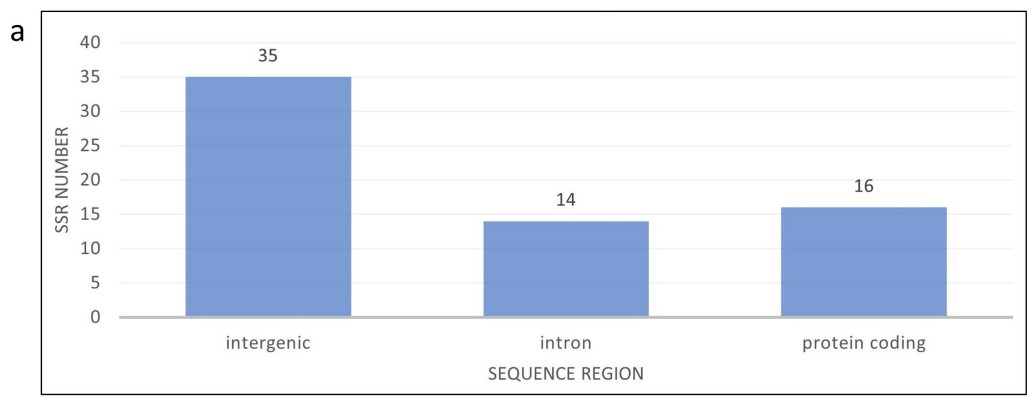

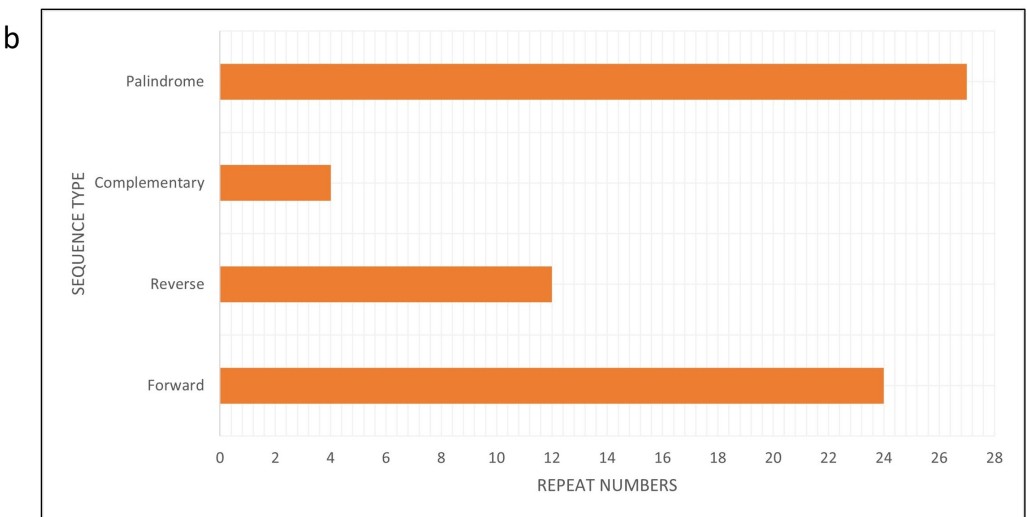

**Figure 4  Bar chart shows SSR distribution in J4-25 chloroplast genome regions.** (A) Bar chart shows SSR distribution in J4-25 chloroplast genome regions. $y$-axis is the SSR number, $x$-axis is the sequence region in the chloroplast region. On the top of each bar is a total SSR number for that region. (B) Horizontal chart analysis of repeated sequences in J4-25 chloroplast genome regions. $y$-axis is the repeat types, $x$-axis is the number of sequences in the chloroplast genome.

forward, reverse and complementary) are present in the chloroplast genome. The results are consistent with repeat types of three other palm family of genus *Euterpe* (*Francisconi et al., 2022*) in which palindrome repeat has the highest number, followed by forward repeat, reverse repeat and finally complementary repeat. This revealed high conservation of gene structure in palm species specifically among genus *Euterpe* and *Elaeis* species.

## CONCLUSIONS

This study has presented the chloroplast genome assembly of 25 *E. guineensis* individuals from short-read paired-end raw data. Maternal inheritance among the $F_1$ progenies is robust with only a few individuals showing a variation in the sequence. Although there are intraspecific variations between male and female plants, they are minimal compared to the reference genome. Genotype evidence shows that *in E. guineensis*, chloroplast genome

sequences are uniparentally inherited through the maternal parent, thus demonstrating their high potential for use to ensure the passing of the desired transgene to the progenies.

## ACKNOWLEDGEMENTS

The authors would like to thank the administrative personnel of Center of Chemical Biology, Universiti Sains Malaysia for the facilities and assistance provided during this study.

### Funding

This work was supported by the Malaysia Research University Network, Long Term Research Grant Scheme (MRUN-LRGS) awarded to Universiti Sains Malaysia (203.PCCB.6777001). The funders had no role in study design, data collection and analysis, decision to publish, or preparation of the manuscript.

### Grant Disclosures

The following grant information was disclosed by the authors:
Malaysia Research University Network, Long Term Research Grant Scheme (MRUN-LRGS): 203.PCCB.6777001.

### Competing Interests

Muhammad Farid Abdul Rahim, Nurul Fatiha Farhana Hanafi & Mohd Azinuddin Ahmad Mokhtar are employees of FGV R&D Sdn Bhd/FGV Holdings, Malaysia. The authors declare there are no competing interests.

### Author Contributions

- Nurul Shakina Mohd Talkah performed the experiments, analyzed the data, prepared figures and/or tables, and approved the final draft.
- Nur Afieqah Kaz Abdul Aziz performed the experiments, prepared figures and/or tables, and approved the final draft.
- Muhammad Farid Abdul Rahim performed the experiments, authored or reviewed drafts of the article, and approved the final draft.
- Nurul Fatiha Farhana Hanafi performed the experiments, authored or reviewed drafts of the article, and approved the final draft.
- Mohd Azinuddin Ahmad Mokhtar conceived and designed the experiments, authored or reviewed drafts of the article, and approved the final draft.
- Ahmad Sofiman Othman conceived and designed the experiments, authored or reviewed drafts of the article, and approved the final draft.

### DNA Deposition

The following information was supplied regarding the deposition of DNA sequences:
All cleaned reads sequences and the complete chloroplast genomes are available at GenBank: SRR24011216 to SRR24011240; OR125032 to OR125036 and OR125038 to OR125056.

## Data Availability

All cleaned reads sequences and the complete chloroplast genomes are available at GenBank: SRR24011216 to SRR24011240; OR125032 to OR125036 and OR125038 to OR125056.

## Supplemental Information

Supplemental information for this article can be found online at http://dx.doi.org/10.7717/peerj.17335#supplemental-information.

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
