# Peer review of "The chloroplast genome inheritance pattern of the Deli-Nigerian prospection material (NPM) × Yangambi population of Elaeis guineensis Jacq"

_PeerJ, doi:10.7717/peerj.17335_

## Round 0.1 · original submission · Minor Revisions

Dear Authors
The manuscript cannot be accepted for publication in its current form. It needs a minor revision to be reconsidered for publication. The authors are invited to revise the paper considering all the suggestions made by the reviewers. Please note that the requested changes are required for publication.
With Thanks

·

Basic reporting

This is a well-written paper on an topic which may be in the interest of a relatively wide range of researchers working in the field. Professional English used throught the text.
The Introduction should be datailed. For example lines 69-82 and 84-94 shoud be improved. These text provide the information concerning common features of plants including angiosperms. Nothing written about investigations concerning genome of Elaeis guineensis Jacq in the Introduction. Here are some literature about E. guineensis genome such as:
- Uthaipaisanwong P. et al. Characterization of the chloroplast genome sequence of oil palm (Elaeis guineensis Jacq.). DOI: 10.1016/j.gene.2012.03.061;
- Jack P.L. et al. Assessment of nuclear, mitochondrial and chloroplast RFLP markers in oil palm (Elaeis guineensis Jacq.). DOI: 10.1007/BF00222128.
The first one cited by authors in Discussion. I think, that this citing is better to place into the Introduction section to elucidate the necessity of their investigation. The aim of the work shoud be described more properly. Is the maternal parent genom investigation is significant in the work? I think that the authors should to explain this in the last paragraph of the Introduction as well.

Experimental design

The experimental work was professionally performed and supported with appropriate and state-of-the-art instrumentation. Tables and figures are rather clear and of the good quality. Methodology is well-described.
Line 109: how leafs were collected? I recommend you to note the sampling station coordinates if it is possible and to mark the date and the season of sampling.
Line 112: I also strongly recommend you to describe procedure of DNA extraction and purification in details.
Line 113: "Genomic DNA quality was evaluated on 1.0% agarose gels..." Please, describe the agarose gel composition more properly.
I recommend you to detalize "Genome assembly and annotations" Issue and to describe the methodology of purification checking.
Line 122: "...any problematic regions..."
Line 150: Haplotype analysis was conducted using DnaSP v6 to.....

"Results' issue well written
Check all abbreviations and interpret them please
InDels
GC content
Figure 2 confuse. Rewrite the capture or modify the figure.

Lines 266-276: the text need additional work

The "Conclusions" are written better than the "Abstract" section.

Validity of the findings

Conclusions are well stated and supported with tables, figures, and the results.
My recomendations to improve the abstract.
Novelty should be marked some brightly.
In a whole the article is interesting and not overloaded.
In my view, the paper can be accepted for publication in PeerJ after minor revision.

Reviewer 2 ·

Basic reporting

The paper aims to explore the genetic pattern among Deli 100 Nigerian Exploration Materials (NPM) x Yangambi, specifically the GB population (J 4101 25 x ML-161). The research methods are described in detail in this paper. However, there are still some errors in the manuscript that need to be corrected.

Experimental design

1.Four types of repeats using the REPuter program are recommended for inclusion in SSR results analysis, including forward (F), reverse (R), Complementary (C), and palindromic (P).
2.Due to the small number of ICONS and the lack of representative charts, it is difficult to explain the genetic pattern of NPM × Yangambi 4 population.

Validity of the findings

no comment

Additional comments

Line 33 It is recommended to change 'analysed' to 'analyzed'.
Line 38It is recommended to change 'genome' to 'genomes'.
Line 40It is recommended to change 'variation' to 'variations'.
Line 75It is recommended to change 'harbours' to 'harbors'.
Line 80It is recommended to change 'include' to 'including'.
Line 80It is recommended to change 'programme' to 'programmes'.
Line 129It is recommended to change 'genome' to 'the genome'.
Line 144It is recommended to change 'insertion' to  'insertions'.
Line 150It is recommended to change 'nexus' to 'the nexus'.
Line 153It is recommended to change 'were' to 'was'.
Line 159It is recommended to change 'comprised of' to 'comprised'.
Line 180It is recommended to change ' coverage' to 'the  coverage'.
Line 185It is recommended to change 'visualisation' to 'visualization'.
Line 188It is recommended to change 'protein coding' to 'protein-coding'.
Line 199It is recommended to change 'mononucleotide' to 'mononucleotides'.
Line 199It is recommended to change 'dinucleotide' to 'dinucleotides'.
Line 199It is recommended to change 'two to' and 'two'.
Line 263It is recommended to change 'considerably' to 'a considerably'.
Line 263It is recommended to change 'the genetic' to 'genetic'.
Line 290It is recommended to change 'Long Term' to 'Long-Term'.
The manuscript lacks a graphic description,and it may be enriched by using the following references: DOI 10.1016/j.scienta.2023.111909; DOI 10.1016/j.scienta.2023.111909

Annotated reviews are not available for download in order to protect the identity of reviewers who chose to remain anonymous.

·

Basic reporting

At first, I would to mention the actual topic of the study – the heredity mechanism of chloroplast of the oil palm. Although, the Intro section barely covers the studies on the topic, in particular what is known about chloroplast inheritance for close related species (not higher plants at whole). I suggest to extend this part with the review of current research. Some studies might be considered:
Yao, G., Zhang, YQ., Barrett, C. et al. A plastid phylogenomic framework for the palm family (Arecaceae). BMC Biol 21, 50 (2023). https://doi.org/10.1186/s12915-023-01544-y
Francisconi AF, Cauz-Santos LA, Morales Marroquín JA, van den Berg C, Alves-Pereira A, Delmondes de Alencar L, et al. (2022) Complete chloroplast genomes and phylogeny in three Euterpe palms (E. edulis, E. oleracea and E. precatoria) from different Brazilian biomes. PLoS ONE 17(7): e0266304. https://doi.org/10.1371/journal.pone.0266304
According to the manuscript the aim of the study is to “clarify inheritance patterns”, but what is known to date about inheritance mechanism among palm species?
Some haplotypes (Table 3) are missing of Figure 2.

Experimental design

Is the sequencing depth of the genome assembly (10X whole-genome coverage for all F1 progenies and 40X for other) sufficient for the complete genome? According to comparison (https://doi.org/10.3389/fpls.2022.779830), the 100× is required and GetOrganelle is the better choice for de novo assembly instead of NOVOPlasty. In the same time authors mention “high confidence level (>100x)” in the Discussion.
What software were used to draw Figure 1? Looks like OGDRAW.

Validity of the findings

The results look interesting and original, but Table 1 is not easy to follow, the key findings might be presented on a figure (f.e. particular genomic regions). Comparison with the reference NC_017602 would be particularly interesting.

---

## Round 0.2 · accepted · Accept

Dear Authors,
I am pleased to inform you that after the last round of revision, the manuscript has been improved a lot, and it can be accepted for publication.

Congratulations on the acceptance of your manuscript and thank you for your interest in submitting your work to PeerJ.

With Thanks

·

Basic reporting

Review (2), 3.04.2024

The Manuscript "The chloroplast genome inheritance pattern of the Deli-Nigerian prospection material (NPM) x Yangambi population of Elaeis guineensis Jacq. significantly" have been significantly improved by authors. Most remarks were taken into account. The careful work have been done. Introduction is well-written and comfortable for understanding the investigation importance. The experimental work is well-described. Tables and figures are of the fine quality. Results issue is well written. I could not find "GC content" abbreviation through the text. Please, add it. In my opinion, the paper is ready for publication in PeerJ.

Experimental design

Introduction is well-written and comfortable for understanding the investigation importance. The experimental work is well-described. Tables and figures are of the fine quality. Results issue is well written.

Validity of the findings

I could not find "GC content" abbreviation through the text. Please, add it.

·

Basic reporting

pass

Experimental design

I suggest to publish the raw sequencing data too for reproduce the results in future by other research groups.

Validity of the findings

pass

Additional comments

none